

# Photosynthesis, yield and raw material quality of sugarcane injured by multiple pests

José A.S. Rossato Jr[1], Leonardo L. Madaleno[2], Márcia J.R. Mutton[3], Leon G. Higley[4] and Odair A. Fernandes[3]

[1] Faculdade Dr. Francisco Maeda—Fafram, Ituverava, SP, Brazil
[2] Paula Souza Center, Technology College, Jaboticabal, São Paulo, Brazil
[3] Universidade Estadual Paulista (Unesp), Faculdade de Ciências Agrárias e Veterinárias, Jaboticabal, SP, Brazil
[4] University of Nebraska—Lincoln, School of Natural Resources, Lincoln, NE, United States of America

## ABSTRACT

Understanding sugarcane (*Saccharum* spp.) response to multiple pest injury, sugarcane borer (*Diatraea saccharalis*) and spittlebug (*Mahanarva fimbriolata*), is essential to make better management decisions. Moreover, the consequences of both pests on the sugarcane raw material quality have not yet been studied. A field experiment was performed in São Paulo State, Brazil, where sugarcane plants were exposed to pests individually or in combination. Plots consisted of a 2-m long row of caged sugarcane plants. Photosynthesis was measured once every 3 months (seasonal measurement). Yield and sugar production were assessed. The measured photosynthesis rate was negatively affected by both borer and spittlebug infestations. Photosynthesis reduction was similar on plants infested by both pests as well as by spittlebug individual infestation. Plants under spittlebug infestation resulted in yield losses and represented 17.6% (individual infestation) and 15.5% (multiple infestations). The sucrose content and the sucrose yield per area were reduced when plants were infested by multiple pests or spittlebug.

## INTRODUCTION

The sugarcane borer, *Diatraea saccharalis* (Fabricius, 1794) (Lepidoptera: Crambidae) is one of the most important pests of sugarcane, *Saccharum* spp., and maize, *Zea mays* L., occurs in several countries of the Americas, and is commonly found in all sugarcane producing areas in those countries (*Dinardo-Miranda, 2008*; *White et al., 2008*). However, sugarcane plants are also injured by several other insect pests that may cause economic losses (*Guagliumi, 1973*).

During the 1990's in Brazil, the use of fire to burn sugarcane fields for manual harvesting was replaced by mechanical harvesting. This new harvesting system allows a large amount of shredded sugarcane leaves and tips to be kept on the soil surface causing environmental changes in sugarcane habitat. Abiotic modifications such as higher soil moisture and lower solar irradiation on the protected surface of the soil have favored outbreaks of the native

Corresponding author
Odair A. Fernandes,
odair.fernandes@unesp.br

spittlebug, *Mahanarva fimbriolata* (Stål, 1854) (Hemiptera: Cercopidae), and this insect has become an important pest of sugarcane in Brazil (*Mendonça, Barbosa & Marques, 1996*; *Dinardo-Miranda, Garcia & Coelho, 2001*; *Garcia et al., 2011*). The spittlebug is particularly important during the wet season when nymphs and adults occur, whereas the sugarcane borer can damage the crop at any time of the year.

Plant mechanisms to reduce stress caused by herbivores are directly and indirectly related to physiologic processes such as respiration, transpiration, and photosynthesis (*Welter, 1989*; *Higley, Browde & Higley, 1993*). Photosynthesis influences plant biomass accumulation, and plants exhibiting high photosynthetic rates may result in higher yields (*Haile, 2001*). In addition to yield reductions, pest injury can also negatively affect yield quality, and reductions in quality have been reported for spittlebug-infested plants (*Madaleno et al., 2008*; *White et al., 2008*; *Ravaneli et al., 2011*).

Because sugarcane production in Brazil now routinely faces simultaneous injury by two important insect pests, characterizing sugarcane responses to these pests individually and in combination is essential for proper pest management. Of particular concern is the recognition that plant response to combined stressors may be greater than the sum of plant response to each pest individually (*Peterson & Higley, 2001*). Despite the importance of the two sugarcane pests and their simultaneous occurrence during part of the growing season, sugarcane response to the injury of these pests combined has not been examined. Moreover, knowledge of plant responses to simultaneous injury by pests can be an important tool to improve current decision-making thresholds, because the occurrence of pests in field may not be isolated in time.

In most agricultural plants, yield (defined by humans as seeds, leaves, tubers, or stems, for example) is indirectly related to primary metabolism. However, in sugarcane yield can be defined as sugar (or a direct product of sugar, like ethanol), and sugar is a direct product of photosynthesis. Consequently, changes in photosynthesis from pest injury should be directly related to yield loss. Thus, our objective in this study was to evaluate the impact of these two pests, individually and in combination, on photosynthesis and its direct impact on yield and yield quality of sugarcane.

## MATERIAL AND METHODS

The experiment was carried out in a commercial sugarcane area (21°19′S and 48°06′W), Ribeirão Preto region, São Paulo State, Brazil in 2008–2009. The sugarcane variety selected was SP80-3280 (4th ratoon), which is considered susceptible to both spittlebug and sugarcane borer (*Dinardo-Miranda, 2003*). Planting density was 12 stalks/row-m.

Typically, studies on photosynthetic responses of plants to insect injury are conducted within a single season (at times of peak photosynthetic activity), while studies on yield losses from insect injury are conducted over multiple seasons (to accommodate environmental variation, particularly, water availability). Because our focus here is on photosynthesis and photosynthate (sugar) accumulation, our experiment was conducted in a single growing season. Additionally, we used this study to establish whether or not multiple pest interactions occurred, which would require more detailed examination. Elsewhere

(*Rossato et al., 2012*), relationships between insect injury and yield loss are reported from multi-seasonal studies arising from the research reported here.

The experimental design was a randomized complete block treatment arrangement with four replications. Treatments were blocked by row. The experimental unit was a 2 × 2 m cage over a row. Experimental units were randomly assigned within a block. Although uncaged controls (a designated 2 × 2 m area in a row) were established, these could not be used for determining potential cage effects, because all uncaged controls had high infestations rates of sugarcane borers. Treatments included caged sugarcane plants a) infested by spittlebug alone; (b) infested by sugarcane borer alone (high infestation); (c) infested by spittlebug + sugarcane borer; and d) plants without insects (control). Consequently, there were three insect treatments and one control treatment for a total of 16 experimental units (4 treatments × 4 blocks). Each experimental unit consisted of 2-m row of sugarcane plants, protected by a metallic cage covered with anti-aphid screen (1 × 1 mm) to avoid insect movement into or out of the experimental units. The cages were placed over plants at the 1 to 2 internode plant growth stage, which is when stalks are naturally infested by sugarcane borer but are without spittlebug infestation.

Although our description of the experimental design implies conventional treatment assignments, this was not the case for the sugarcane borer treatments. Our original plan was to artificially infest cages with sugarcane borers; however, plants were infested before exclusion cages could be used. Consequently, all treatments (with and without sugarcane borers) had to be established by inspecting plants for evidence of infestation. Establishing treatments in this way seems to violate the assumption of random treatment assignment.

The principle of random treatment assignment is used to avoid potential confounding with treatment effects. In other words, the only factor influencing an experimental unit should be the treatment (or treatment and block effect in blocked designs). Our choice of experimental units is within a sugarcane planting in which plants have the same genetics and are grown under uniform agronomic conditions. While location effects on experimental units were possible, these were accommodated through blocking. Consequently, the one potential confounding factor in assigning treatments based on pre-existing infestations, is whether or not the infestations themselves were random or non-random. Studies on sugarcane borer spatial variability in sugarcane have established that infestations have an aggregated distribution, with areas of higher infestation density randomly distributed through a field (*Anjos et al., 2010*). Given this understanding of sugarcane borer biology, we believe the use of pre-existing infestation sites for treatment assignment is consistent with the notion of random treatment assignment, albeit with the sugarcane borers doing the randomization rather than the experimenters!

A final point on sugarcane borer treatments is that because infestations have a random distribution, within a group of plants (i.e., within an experimental unit), it is possible to have high and low levels of infestation among and within plants. Infestation levels of sugarcane borers are determined destructively by splitting stalks, counting sugarcane borer tunnels, and measuring tunnel lengths. Because this approach was not useful for assigning treatments, we used visual inspections for frass (larval feces) and entry holes as indicators. For sugarcane borer-free experimental units we chose plants without frass or entry holes.

For sugarcane borer-infested experimental units we chose plants subjectively with high and low levels of infestation indicators. For the sugarcane borer plus spittlebug treatment we identified (again subjectively) high infestation levels of sugarcane borers. Differences in actual sugarcane borer infestation levels were determined at the end of the experiment by splitting stalks and measuring tunneling.

Spittlebugs nymphs were used to artificially infest treatments. Nymphs hatched from diapausing eggs and collected from the field. After infestation, spittlebug nymphs were monitored on every stalk at 2 to 3-day intervals and counted. Nymphs were removed from or added to the cages to keep similar infestations in the spittlebug-infested plots. The spittlebug infestation density was expressed as daily infestation (nymph/m/day) as suggested by *Madaleno et al. (2008)*.

Photosynthetic rates of sugarcane plants were measured on the leaf +3 (*Van Dillewijn, 1952*) of three plants in each plot. A portable photosynthetic system (Li-Cor, Model LI-6400) was used in each of the seasons: in February (121 days after plant emergence), April (170 days after plant emergence), June (254 days after plant emergence), and September (346 days after plant emergence), which characterized summer, fall, winter, and spring, respectively. Measurements were made at an average (ambient) $CO_2$ concentration of 409 ppm, average light intensity of 900 µE PAR (photosynthetically active radiation) from LED blue-red light source, and average leaf temperature of 25.3 °C. All photosynthetic measurements were made until the CV for photosynthesis was less than 3%.

During harvest, all senescent leaves were stripped out and all stalks were cut manually from each plot. Length and diameter of each stalk was measured at harvest (348 days after plant emergence). The diameter of each stalk was measured at the middle of the lower most, middle, and upper most internodes using a handheld pachymeter. Stalk yield was obtained using the formula: $[(\text{diameter}^2 {*} 15 {*} \text{height} {*} 0.007854)/1.5]$, as described by *Landell & Bressiani (2008)*. The total internodes were counted. Stalks were longitudinally split to evaluate the internodes injured by sugarcane borer and to determine the Infestation Intensity (II) by dividing the number of borer-damaged internodes by the total number of internodes.

All stalks within a plot were shredded and homogenized to extract the sugarcane juice by a hydraulic press (*Tanimoto, 1964*). Immediately after extraction, the level of soluble solids (Brix) and the apparent sucrose content (Pol) were determined using methods proposed by *Schneider (1979)*. Phenolic compounds were estimated as proposed by *Folin & Ciocalteu (1927)*. The sucrose yield per area was calculated using the sucrose content (Pol) and the stalk yield.

Data were analyzed with SAS (SAS Institute, Cary, NC), and treatment effects were determined by repeated measures analysis of variance (GLM). For means comparisons, we used an unprotected least significant difference (LSD) with ($P \leq 0.05$). We chose the unprotected LSD at $P > 0.05$ in evaluating differences among means for three reasons. First, our core interest is in identifying differences of insect treatments from the control, and even in a non-significant ANOVA the control could be significantly different from some insect treatments. Second, because we have few treatments the issue of generating false significance is reduced. And third, we wanted to use a liberal comparison so any

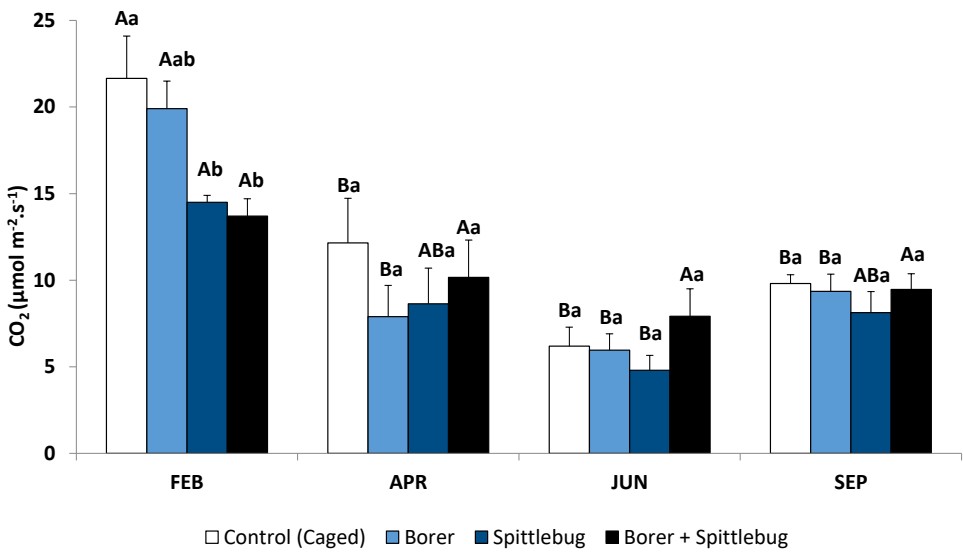

**Figure 1** **Photosynthetic rates of sugarcane plants infested by sugarcane borer, spittlebug, and combined pests.** Means followed by different letters, capital letters comparing seasons and small letters comparing treatments, were significantly different by LSD Test ($P \leq 0.05$).

potential biological differences among treatments would be identified (or stated another way, we chose to reduce alpha error at the expense of increasing beta error).

Graphic presentations of photosynthesis-yield relationships and all regressions were developed with Prism 6.01 software (GraphPad Software, Inc., La Jolla, CA, USA; http://www.graphpad.com/scientific-software/prism/). Details on methods used for evaluating fit of regression models are available in GraphPad documentation (GraphPad Curve Fitting Guide at http://www.graphpad.com/guides/prism/6/curve-fitting/index.htm).

## RESULTS AND DISCUSSION

The mean infestation of sugarcane borer was 15.8% which is considered a high sugarcane borer infestation. In the spittlebug treatment, the mean number of nymphs was 3.07 nymphs/m/day. In the treatment with both pests combined, the infestation intensity of sugarcane borer was 13.63% and 2.95 nymphs/m/day for spittlebug, which were similar to the infestations observed in sugarcane borer and spittlebug treatments, respectively. The control (represented by non-infested caged plants) was not infested by the pests.

Sugarcane plants had the highest average photosynthetic rates in the summer (February) with photosynthetic rates in other seasons approximately half of those in the summer (Fig. 1, control treatments). Abiotic factors such temperature, net radiation, and especially rainfall (Fig. 2), are most favorable to plant development during the summer and likely to account for these photosynthetic rate differences (*Taiz & Zeiger, 2006*).

Reductions in photosynthesis were noted from insect injury, but reductions varied by insect and by season. Generally, reductions were observed depending on the stressor on
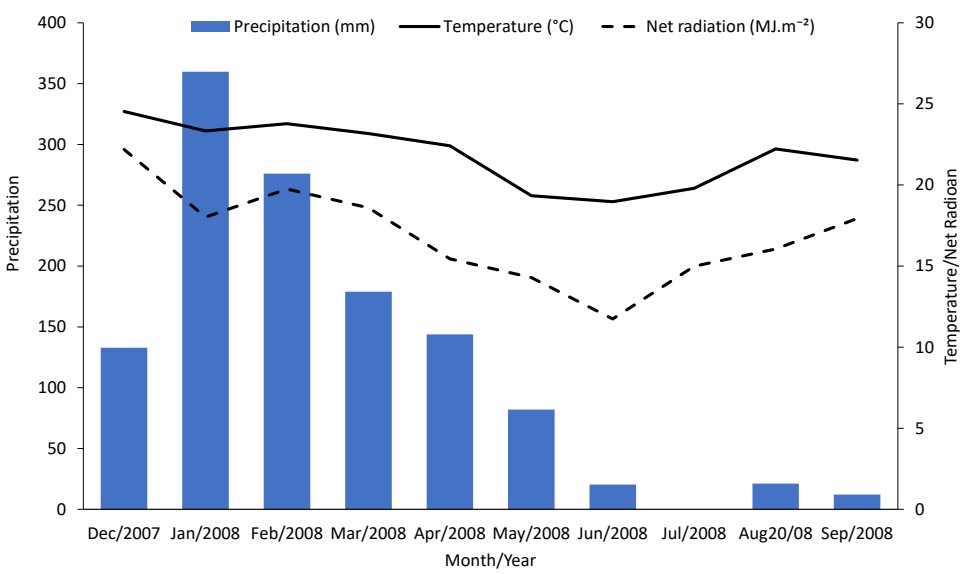

**Figure 2** Monthly accumulated rainfall precipitation and mean temperature and net radiation measured in the experimental area.

the plants. Sugarcane plants infested with spittlebugs (alone or combined with borer) had lower photosynthetic rates. Therefore, spittlebug injury reduced photosynthesis.

During the period of spittlebug infestation (summer), sugarcane plants showed significant photosynthesis reduction (Fig. 1, Table 1). However, there were no differences between plants under spittlebug infestation and combined infestation of spittlebug and borer, even though it is known that galleries caused by the sugarcane borers can reduce water flow within the plants. This result suggests that the spittlebug injury is worse than borer injury for sugarcane plants. Probably, this impact is related to the long period of feeding by nymphs on the roots. Damage caused by sucking insects may vary considerably depending on the time length of feeding (*Reddall et al., 2004*; *Gomez et al., 2006*). Spittlebug nymphs may interrupt the phloem vessels to the roots causing root death (*Garcia, Botelho & Parra, 2006*). Also, sucking insects, in general, may remove plant tissue affecting physiological processes, release saliva that is toxic to the plants, and cause tissue necrosis (*Fewkes, 1969*; *Haile, 2001*). Nevertheless, when the spittlebug infestation ended in April photosynthetic rate differences were no longer observed.

Reductions in photosynthesis from sugarcane borer were statistically significant in April, June, and September. Injury from sugarcane borer is usually greater (longer galleries in the stalk) in the fall than summer (*Macedo & Botelho, 1988*), and this may have negatively influenced the physiological processes during this season. Comparing these two seasons, plants with sugarcane borer infestation had showed 63% (fall) and 56.8% (summer) photosynthetic reductions whereas non-infested plants showed 43.9%. Considering that in April the plants were not under optimal abiotic factors (Fig. 2), these data suggest that the difference between photosynthetic rates between non-infested plants and plants under sugarcane borer infestation represents the negative impact of the borer on the plants. The

galleries in the stalks may promote similar stress in the plant as well as drought conditions. Therefore, plants under both abiotic and biotic stressors may have a reduction in nutrients and water flow to leaves (*Culy, 2001*) and result in a decrease of accumulated biomass (*Vaadia, 1985*).

The effect of photosynthetic rate reduction was reflected on yield at harvest. Plants with spittlebug infestation (individually or combined) showed thinner stalks, and some plants completely dried. Similar results were also reported by *Dinardo-Miranda (2003)*. In this study the diameter and length of stalks were significantly affected when spittlebug was present. This impact in the stalks was caused by the spittlebug nymphs whose feeding injures the roots affecting phloem and xylem flow of water and nutrients, such as nitrogen, phosphorus, potassium, calcium, and glucose (*Garcia, Botelho & Parra, 2006*; *Dinardo-Miranda, 2008*).

On the other hand, the diameter of the stalk was not affected by sugarcane borer infestation (Table 1). The infestation intensity was not sufficient to cause any stalk diameter reduction, despite photosynthetic rate reduction observed in April. Diameter and length reductions were enough to cause sugarcane yield losses. Plants infested by spittlebug (alone or combined) were affected negatively for yield losses. Compared to the uninfested plants, stalk yield reduction was 17.6 and 15.5% under spittlebug individually or spittlebug combined with borer, respectively.

According to *White et al. (2008)*, stalk yield losses are positively correlated to borer infestation intensity. However, despite the infestation intensity observed (15.8%), there was no significant stalk yield reduction. It is possible that some sugarcane varieties may have mechanisms to prevent yield losses even under such an infestation intensity of sugarcane borer. Moreover, the current methodology to estimate injury based on infestation intensities may not predict the actual injury caused by borer in the stalk. Therefore, studies involving the volume of gallery (length and diameter of the tunnels) made by borers may better represent the infestation ($\sim$ injury) instead of using bored internodes as an infestation intensity parameter because the severity of injury is partially assessed.

There was no difference in the levels of soluble solids in the raw material obtained from plants infested with borer and/or spittlebug. Similar results were observed on plants with spittlebug infestation by *Garcia et al. (2010)* and *Ravaneli et al. (2011)*.

The sucrose yield per area was negatively affected by spittlebug injury, decreasing 15.1 and 16.6%, individually or combined with borers, respectively. Regardless of the infestation level of sugarcane borer, there was no significant difference in sucrose yield per area. Thus, these results confirmed that spittlebug injury impact (alone or combined) is worse than borer injury for sucrose yield per area, which is most likely influenced by the stalk reduction.

Typically, borer infestations are associated with opportunistic fungi *Fusarium moliniforme* and *Colletotrichum falcatum* infections. These pathogens enter the galleries and induce the production of metabolite inhibitors and lead to sucrose inversion (*Ingram, 1946*; *Stupiello, 2010*). However, the amount of phenolic compounds was not affected by these fungi, even with pest infestation (Table 1). Sugarcane plants exposed to greater pest infestations may increase the amount of these phenolic compounds which affect quantitatively and qualitatively both sugar and ethanol productions (*Ravaneli et al., 2011*).

Rossato Jr. et al. (2019), *PeerJ*, DOI 10.7717/peerj.6166

**Table 1  Descriptive statistics (means and standard errors in parentheses), and ANOVA for biometric parameters at sugarcane harvest, including: stalk diameter, length, and wet weight (per ha); sucrose (per ha), phenolics (density per stalk), Brix (%), starch concentration (per stalk), and brix (%).** Treatments included infestation by the sugarcane borer (*Diatraea saccharalis*) and spittlebug (*Mahanarva fimbriolata*), both pests, and a caged control. Letters after the mean (SE) indicate significant treatment differences by unprotected LSD.

| Treatment | Stalk diameter (cm) | Stalk Length (cm) | Stalk Fresh Weight (t ha$^{-1}$) | Sucrose (t ha$^{-1}$) | Phenolics (mg dm$^{-3}$) | Starch (mg ml$^{-1}$) | Brix (%) |
|---|---|---|---|---|---|---|---|
| Control | 2.49 (0.04)a | 272.82 (3.08)a | 125.54 (3.43)a | 24.50 (0.95)a | 322.11 (11.07)a | 215.41 (14.38)a | 21.74 (0.26)a |
| Borer | 2.43 (0.01)ab | 258.63 (7.34)ab | 116.85 (4.73)ab | 22.96 (1.06)ab | 348.42 (38.62)a | 130.78 (37.21)a | 21.93 (0.18)a |
| Spittlebug | 2.32 (0.04)b | 245.44 (9.58)b | 103.41 (5.00)b | 20.79 (1.15)b | 398.22 (46.23)a | 222.65 (35.16)a | 22.22 (0.18)a |
| Combination (Borer + Spittlebug) | 2.36 (0.02)b | 248.47 (3.18)b | 106.04 (5.16)b | 20.42 (0.86)b | 362.79 (51.02)a | 156.40 (32.69)a | 21.79 (0.39)a |
| | | | **ANOVA Treatment Effects** | | | | |
| $F_{3,9}$ | 4.46 | 3.31 | 5.70 | 4.67 | 1.06 | 2.03 | 1.12 |
| $Pr > F$ | 0.035 | 0.071 | 0.018 | 0.031 | 0.414 | 0.181 | 0.390 |
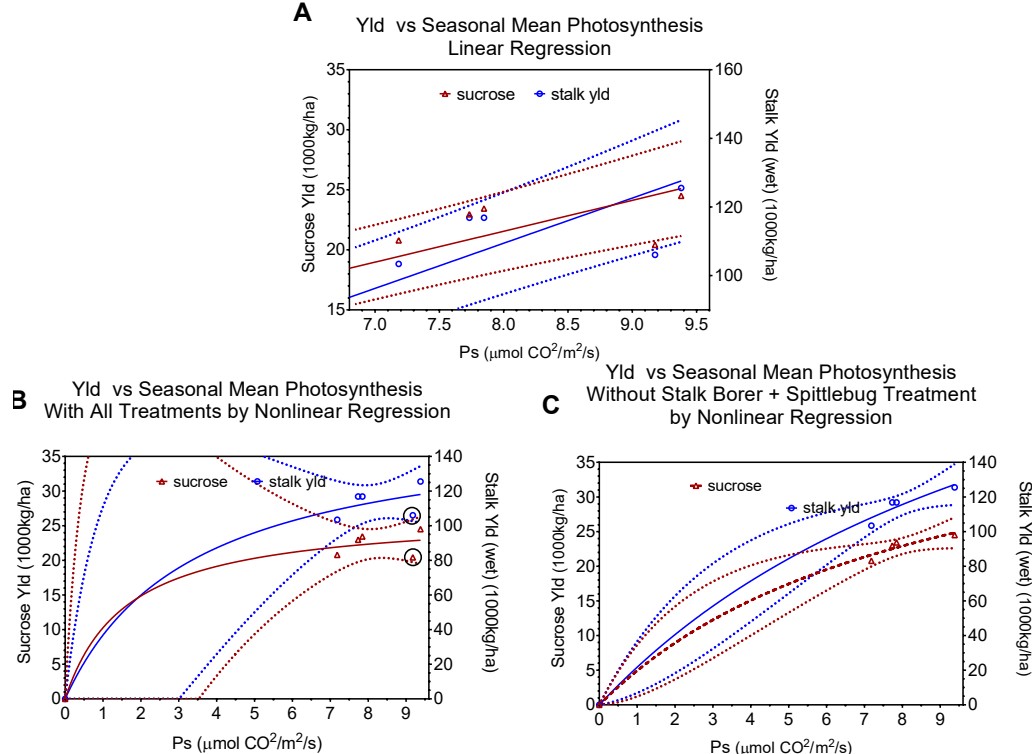

**Figure 3** Regression relationships between seasonal mean photosynthesis and final sucrose yield and stalk weight. Dotted lines indicate 95% confidence intervals for the regressions. (A) shows the linear regression of yield versus seasonal mean photosynthesis and includes all data points. (B) shows the non-linear ($Y = Vmax * X/(Km + X)$) fit with all data points (points for combined injury are circled in black). (C) shows the non-linear $Y = Vmax * X/(Km + X)$) fit excluding points for combined injury. Regression statistics are listed in Table 2.

In contrast to the results obtained by *Garcia et al. (2010)* and *Ravaneli et al. (2011)*, the pest infestation was probably lower in our study. Moreover, sugarcane plant produces phenolic compounds naturally and the amount produced is also affected by the plant age and variety (*Simioni et al., 2006*). In addition to examining individual treatments, we were interested in potential changes between parameters, particularly between photosynthesis and yield. Although total photosynthesis clearly relates to yield capacity, efforts to directly relate measures of leaf photosynthesis to final yield have not been very successful (*Long et al., 2006*) largely because of the many factors (environmental and genetic) that influence this relationship. In looking at insect injury to sugarcane, our hope was that photosynthesis and sugar accumulation would offer a more direct relationship than carbon accumulation and photosynthesis in other plant systems.

We used regressions and graphs to illustrate photosynthesis and yield (Fig. 3, Table 2). Figure 3A shows a simple linear regression of yield (sucrose or total stalk wet weight) versus photosynthesis, with data points representing the average by treatment. Our intent with this regression was not to produce a realistic model of the photosynthesis-yield relationship, but rather to see if photosynthesis and yield showed a positive association, independent
**Table 2  Regression statistics for regressions reported in Figs. 3 and 4 : yield (sucrose or stem weight) vs. apparent photosynthesis rates.** For all linear regressions *df* for *F* is 1,3 except for mean seasonal photosynthesis which is 1, 4. Non-linear regressions (Figs. 3B and 3C) are ''Michaelis–Menten'' equations of the form $Y = V_{max}*X/(K_m + X)$; goodness of fit was evaluated by $R^2$ and runs test (no significant deviation from model was found for any non-linear regression).

| Figure | Photosynthesis (= X) | Yield (=Y) | Equation | $R^2$ | $F$ | $P > F$ |
|---|---|---|---|---|---|---|
| 3A | Mean Seasonal | Sucrose | $Y = 2.573*X + 0.9656$ | 0.9305 | 53.52 | 0.0019 |
| | | Stalk Weight | $Y = 13.16*X + 4.158$ | 0.9452 | 68.99 | 0.0011 |
| 3B | Mean Seasonal with combined injury | Sucrose | $Y = 26.84*X/(1.613 + X)$ | 0.9732 | | |
| | | Stalk Weight | $Y = 160.20*X/(13.49+ X)$ | 0.9761 | | |
| 3C | Mean Seasonal without combined injury | Sucrose | $Y = 47.95*X/(8.731 + X)$ | 0.9963 | | |
| | | Stalk Weight | $Y = 302.00*X/(12.89 + X)$ | 0.9960 | | |
| 4A | February | Sucrose | $Y = 0.4518*X + 14.17$ | 0.9309 | 26.95 | 0.0352 |
| | | Stalk Weight | $Y = 2.558*X + 66.13$ | 0.8704 | 13.43 | 0.0671 |
| 4B | April | Sucrose | $Y = 0.4975*X + 18.29$ | 0.3720 | 1.185 | NS |
| | | Stalk Weight | $Y = 3.062*X + 87.14$ | 0.4110 | 1.396 | NS |
| 4C | June | Sucrose | $Y = 2.111*X + 10.67$ | 0.8750 | 13.99 | 0.0646 |
| | | Stalk Weight | $Y = 11.84*X + 46.93$ | 0.8029 | 8.147 | 0.104 |
| 4D | September | Sucrose | $Y = 1.804*X + 6.710$ | 0.7400 | 5.691 | NS |
| | | Stalk Weight | $Y = 11.32*X + 13.92$ | 0.8499 | 11.32 | 0.0781 |

of other treatment effects. As expected the patterns for sucrose and total stalk weights are similar, but the pattern of increasing yield with increasing mean leaf photosynthesis is not. The points at a mean photosynthetic rate of ca. 9.3 are noticeable lower than expected, and these points are from the sugarcane borer + spittlebug treatment. Figures 3A and 3B use the same data points in non-linear regressions of a Michaelis–Menten equation to offer a more realistic description of relationship between mean photosynthesis and yield. Although Figs. 3A and 3B present curves showing increasing photosynthesis leading to increasing yield (asymptotically), it is noteworthy that excluding the stalk-borer + spittlebug treatment substantially reduces the confidence intervals (Fig. 3B).

The physiological implications of injury from the combination of stalk-borer + spittlebug are more evident when we look at data by month (i.e.; season), rather than as an average across months (Fig. 4, Table 2). In February (Fig. 4A), the combined treatment has the lowest photosynthetic rates (ca. 13.8) of any treatment. However, in April (Fig. 4B), June (Fig. 4C), and September (Fig. 4D) the combined insect treatment has among the highest photosynthetic rates despite having the lowest associated yields. Consequently, data from April, June, and September for the combined insect treatment is both in contrast to theoretical expectations for photosynthesis and yield and in contrast to relationships seen in other treatments.

Theoretically, the potential interaction between pests can involve two mechanisms: (1) stress incidence interactions, where the occurrence of one stressor increases or decreases the incidence of a different stressor, and (2) stress response interactions, where the occurrence of multiple stressors ''indicate that physiological processes affected by the stresses are interrelated with respect to a measure of damage'' (*Higley, Browde & Higley, 1993*). Because

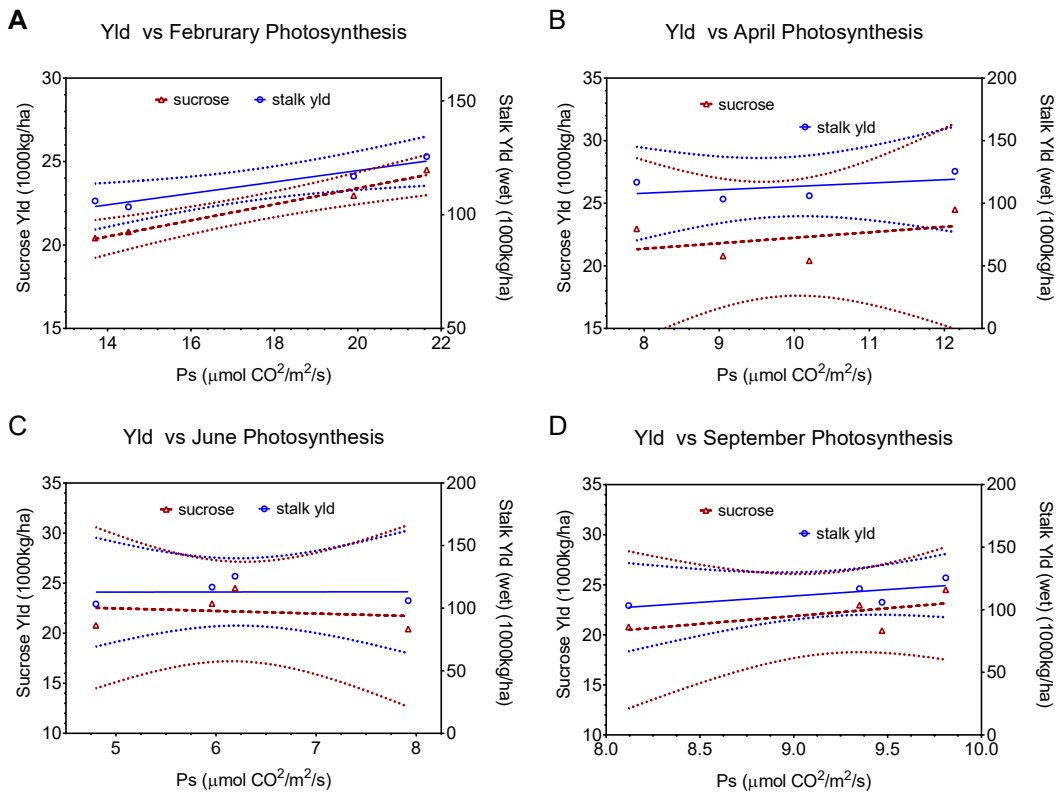

**Figure 4** **Linear regression relationships between photosynthesis and final sucrose yield and stalk weight by month (A February, B April, C June, and D September). Dotted lines indicate 95% confidence intervals for the regressions.** Regressions include all points except the combined (sugarcane borer+ spittlebug) treatments. Regression statistics are listed in Table 2.

the incidence of stressors (spittlebugs and stalk borers) was set experimentally, a stress response interaction, a physiological change, must underlie our observed results.

Fundamentally, there are two possible explanations for our observed pattern of photosynthetic activity and yield. First, if yield potential was set in February (early in sugarcane growth), then the combined treatment would have low yields, because the treatment had low photosynthetic rates in February. However, we see a continued positive association between leaf photosynthesis and final yield in all other treatments after February, which argues against determination of yield potential earlier. Moreover, yield in sugarcane is associated with vegetative growth not reproductive growth (which is where regulation of yield potential is most likely). Finally, sucrose accumulation in this system occurs between April and September, when the combined treatment had high photosynthetic rates, yet had low sucrose accumulation.

Alternatively, the combined insect injury might reflect an uncoupling of photosynthesis with photosynthate accumulation or of a dramatic increase in costs of photosynthate accumulation. Under normal conditions, photosynthetic rates are tightly regulated in association with sink demand, as moderated by environmental conditions (especially temperature and water availability). It is difficult to see how photosynthesis would

be uncoupled from photosynthate production without producing down regulation of photosynthetic enzymes and(or), through the production of peroxides, causing significant damage to thylakoid membranes leading to visible symptoms like chlorosis (e.g., *Macedo & Botelho, 1988*).

Because injury from both insects influences water use and availability, impaired water relations is one potential mechanism by which photosynthesis and sugar accumulation might be disrupted. Ordinarily, significant limitations in water availability are associated with reduced stomatal conductance and (depending on the severity of the water stress) reduced photosynthesis. We found no evidence of either effect (reduced stomatal conductance or reduced photosynthesis) in measurements in April, June, or September; on the contrary, conductance and photosynthesis were higher with combined injury, as compared to other treatments.

However, effects of injury on other aspects of sugarcane metabolism and water are possible. Accumulation of sucrose requires movement against an osmotic gradient, consequently there is an energy cost for this process. If injury of the combined insects impedes this process, perhaps increased costs are required for sugar accumulation. Under such circumstances, there would not necessarily be down regulation of photosynthesis despite reduced sugar mobilization (through end product inhibition), because extra energy would be needed to maintain sugar mobilization. Consequently, if injury by the combination of sugarcane borer + spittlebug leads to increased costs for sugar accumulation, these costs might be indicated by higher respiration rates. With our data, this explanation seems the most likely in accounting for the observed effects.

Beyond the interesting physiological issues raised by our examination of sugarcane responses to multiple insect injuries, we see two other important outcomes from this study. First, our work illustrates that, as we originally hypothesized, sugarcane provides an excellent system for examining the relationship between primary metabolism and yield (expressed as either sucrose or total dry matter accumulation) unlike most other plant systems. Second, looking at how individual and combined insect herbivores alter primary metabolism as well as yield (however defined) can reveal relationships and interactions that are otherwise hidden.

## ACKNOWLEDGEMENTS

The authors are grateful to São Martinho Sugar Mill for the technical support provided and Dr. Jessica Jurzenski for the English review of the first draft.

### Funding

This study was financed in part by the Coordenação de Aperfeiçoamento de Pessoal de Nível Superior - Brasil (CAPES) - Finance Code 001. The funders had no role in study design, data collection and analysis, decision to publish, or preparation of the manuscript.

## Grant Disclosures

The following grant information was disclosed by the authors:

Coordenação de Aperfeiçoamento de Pessoal de Nível Superior - Brasil (CAPES).

## Competing Interests

Leon G. Higley was an Academic Editor for PeerJ.

## Author Contributions

- José A.S. Rossato Jr. conceived and designed the experiments, performed the experiments, analyzed the data, prepared figures and/or tables, authored or reviewed drafts of the paper, approved the final draft.
- Leonardo L. Madaleno performed the experiments, analyzed the data, authored or reviewed drafts of the paper.
- Márcia J.R. Mutton conceived and designed the experiments, analyzed the data, contributed reagents/materials/analysis tools, authored or reviewed drafts of the paper.
- Leon G. Higley conceived and designed the experiments, analyzed the data, prepared figures and/or tables, authored or reviewed drafts of the paper, approved the final draft.
- Odair A. Fernandes conceived and designed the experiments, analyzed the data, contributed reagents/materials/analysis tools, prepared figures and/or tables, authored or reviewed drafts of the paper, approved the final draft.

## Data Availability

The raw data are provided in the Supplemental File.

## Supplemental Information

Supplemental information for this article can be found online at http://dx.doi.org/10.7717/peerj.6166#supplemental-information.

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
