# Peer review of "Photosynthesis, yield and raw material quality of sugarcane injured by multiple pests"

_PeerJ, doi:10.7717/peerj.6166_

## Round 0.1 · original submission · Major Revisions

The first reviewer has made important points about the design and replication of the experiments, especially that another replication is needed to provide confidence that the results are robust.

·

Basic reporting

The manuscript provides information on the physiological, yield, and quality of sugarcane in response to injury by two insect species exhibiting different injury types. The information fits into the broader field of knowledge. Relevant literature is appropriately referenced. The paper is fairly well written but there are needed changes to syntax, style, and grammar.

Experimental design

The experimental and treatment design needs to be explained in more detail. What year was the experiment conducted? Was the experiment replicated just once? If so, why? Field experiments typically are replicated twice to control for the role of environment in influencing any treatment differences. A RCBD design was used, but what was the blocking factor? The authors mention they used ANOVA and an LSD Test. Why? What advantages do those procedures have over others? The authors expect to see differences among treatments so why are they using a relatively liberal means separation test? This might bias their findings. Why not use a more conservative test instead (e.g., Tukey’s, SNK)? What statistical program was used? R, SAS, other? The authors seem to have measured photosynthesis from the same experimental units throughout the seasons yet there is no mention of a repeated measures design or analysis. Why not? Also, why isn’t there a statistical evaluation for interaction between sugarcane borer and spittlebug? This seems like a highly relevant and desirable question. The authors recorded CO2 exchange rates (photosynthetic rates) but make no mention of other gas exchange variables that can be measured by the LI-6400, like stomatal conductance, intercellular CO2, and transpiration rates. These values could help tease apart potential mechanisms of impairment for spittlebug.

Validity of the findings

See above.

Additional comments

Title: Herbivores do not attack plants. Do cows attack grass? No, so why is it acceptable to say that sugarcane borer attacks sugarcane? Herbivores injure or feed on plants. Change title to: “Photosynthesis, yield, and raw material quality of sugarcane injured by Diatraea saccharalis and Mahanarva fimbriolata. Change “attack” to “injury/injure, etc” throughout the article.
Line 28. Replace “most” with “more”. Include species names with authorities for sugarcane and maize at first mention.
L 29. Insert “in those countries” after “areas”.
L 33. Change “harvest” to “harvesting”.
L 35. Delete “the”.
L 37. Insert “native” before “spittlebug”.
L 38. Delete “native”.
L 42. Delete “the”.
L 44. Delete “the”.
L 45. Replace “present superior” with “result in high yields”.
L 47. Need to define “raw material quality”.
L 52. Replace “(biotic stressors) was not addressed yet” with “have not been characterized”.
L 53. Replace “on” with “of”.
L 54. Delete the phrase after “thresholds”.
L 60. Delete “pests”.
L 61. Replace “It was adopted the…” with “A randomized complete block design (RCBD) with four replications was used.”
L 66. Replace “comprised” with “was”.
L 66. What was the density of the sugarcane plants within the 2-m long rows?
L 67. What are the dimensions of the cage and what is the size of the screening? How long were the cages on the plants?
L 70. Replace “presented” with “present”.
L 77. What is meant by “leaf +1”?
L 78. Need much more information on the LI-COR 6400. Location of manufacturer (Lincoln, Nebraska, USA). Did the authors use the integrated light source or an external source (i.e., sunlight)? What was the light intensity? Was a constant CO2 reference used? If so what was the concentration? What was the flow rate?
L 83. Replace “presented” with “had”.
L 84. Replace “photosntesis” with “photosynthesis”.
L 88. Replace “harvesting” with “harvest”.
L 93. Replace “damaged” with “injured”.
L 105. Replace “15.80%” with “15.8%”.
L 106-107. Very awkward phrasing. Rewrite.
L 109. Replace “close” with “similar”.
L 109. Insert “the” after “in”.
L 111. Replace “presented borer infestation” with “were infested”.
L 112. Replace “had the photosynthetic rate analyzed” with “were analyzed for photosynthesis.”
L 113. Replace “not enough” with “of insufficient number”.
L 115. Insert “average” before “photosynthetic” and delete “average” after “rate”. Replace “on” with “in”.
L 118. Replace “performance” with “result”.
L 120. Insert “sugarcane” before “borer” and replace “presented” with “had”.
L 121. Replace “interfere” with “affect”.
L 126. Delete “the” and replace “feeding” with “injury”.
L 130. Replace “sap” with “phloem”.
Figure 1. Need “CO2” in the y-axis.
L 140. Delete both instances of “the”.
L 141. Replace “higher” with “greater”.
L 143. Insert “negatively” in the sentence before “influenced”.
L 148. Replace “into” with “in”.
Figure 2. The “-2” before “m” needs to be superscripted.
L 155. Delete the second “the”.
L 156. Replace “being that” with “and”
L 158. Replace “were” with “was”.
L 160. Replace “damages” with “injures”.
L 162. Replace “On the other hand” with “However”.
L 162. Insert “the” before “stalk”.
L 163. Delete “occurred” and replace “enough” with “sufficient”.
L 164-165. Delete “Biometric parameters impacts, such as” and begin sentence with “Diameter…”
L 165. Delete comma after “reductions” and replace “enough” with “sufficient”.
L 165-166. Awkward sentence. Rewrite.
L 170. Replace “registered” with “observed”.
L 172 and elsewhere. The “II” abbreviation is not necessary in this manuscript. Just spell out the term each time.
L 185. Replace “Usually” with “Typically” and “to” with “with”.
L 186. Delete “into”.
L 197. Replace “around” with “on”.
L 199. Delete “holistic”.

·

Basic reporting

Authors have done a very good job of reporting their research results.

Experimental design

Trials were very well designed. Authors have really prepared a thorough treatment list to ensure that effects of caging were factored in so that treatment effects were better measured.

Validity of the findings

Results of thier finding adds to our growing knowledge to better understand how such criptic insects such as the sugarcane borer and spittlebugs affect various plant physiological processes, sugar content, and yield of sugarcane. To my knowledge, this is the first work to addres physiological responses of sugarcane to insect feeding and how these ultimatley affects yield, Such knowledge is crucial to understnd the relationship between insect feeding, pest pressure, and threshold levels, which are key to developing managment decisions.

Additional comments

comments are made directly on the manuscript.

---

## Round 0.2 · accepted · Accept

The authors have addressed all the concerns raised by the reviewers except the fact that the experiment was done only during one season. The authors seem to have justified why it was only done in one season. Even though it is not entirely convincing, I am willing to accept the article as it is, due to the time that would be needed to repeat the experiment.

#